# Use of Antiplatelet Agents and Survival of Tuberculosis Patients: A Population-Based Cohort Study

**DOI:** 10.3390/jcm8070923

**Published:** 2019-06-27

**Authors:** Meng-Rui Lee, Ming-Chia Lee, Chia-Hao Chang, Chia-Jung Liu, Lih-Yu Chang, Jun-Fu Zhang, Jann-Yuan Wang, Chih-Hsin Lee

**Affiliations:** 1Department of Internal Medicine, National Taiwan University Hospital, Hsin-Chu Branch, Hsinchu 30059, Taiwan; 2Department of Internal Medicine, National Taiwan University Hospital, Taipei 10002, Taiwan; 3Institute of Epidemiology and Preventive Medicine, College of Public Health, National Taiwan University, Taipei 10052, Taiwan; 4Department of Pharmacy, New Taipei City Hospital, New Taipei City 24141, Taiwan; 5School of Pharmacy, College of Pharmacy, Taipei Medical University, Taipei 11031, Taiwan; 6Department of Internal Medicine, School of Medicine, College of Medicine, Taipei Medical University, Taipei 11031, Taiwan; 7Pulmonary Research Center, Division of Pulmonary Medicine, Wan Fang Hospital, Taipei Medical University, Taipei 11696, Taiwan

**Keywords:** tuberculosis, antiplatelet, aspirin, immunomodulation, survival, Taiwan

## Abstract

While evidence is accumulating that platelets contribute to tissue destruction in tuberculosis (TB) disease, it is still not known whether antiplatelet agents are beneficial to TB patients. We performed this retrospective cohort study and identified incident TB cases in the Taiwan National Tuberculosis Registry from 2008 to 2014. These cases were further classified into antiplatelet users and non-users according to the use of antiplatelet agents prior to the TB diagnosis, and the cohorts were matched using propensity scores (PSs). The primary outcome was survival after a TB diagnosis. In total, 74,753 incident TB cases were recruited; 9497 (12.7%) were antiplatelet users, and 7764 (10.4%) were aspirin (ASA) users. A 1:1 PS-matched cohort with 8864 antiplatelet agent users and 8864 non-users was created. After PS matching, antiplatelet use remained associated with a longer survival (adjusted hazard ratio (HR): 0.91, 95% confidence interval (CI): 0.88–0.95, *p* < 0.0001). The risk of major bleeding was not elevated in antiplatelet users compared to non-users (*p* = 0.604). This study shows that use of antiplatelet agents has been associated with improved survival in TB patients. The immunomodulatory and anti-inflammatory effects of antiplatelet agents in TB disease warrant further investigation. Antiplatelets are promising as an adjunct anti-TB therapy.

## 1. Introduction

Tuberculosis (TB) remains an important global infectious disease with an estimated 10 million new cases and 1.6 million deaths in 2017 [1]. The World Health Organization (WHO) has set a goal of eradicating TB as a public health problem, aiming to achieve 50% and 90% reductions in the global TB incidence by 2025 and 2035, respectively [2]. TB remains an important infectious disease that causes significant morbidity and mortality [3]. Though considered a treatable infectious disease, TB treatment success rates and cure rates are still suboptimal [4]. The mortality rate from TB remains high, especially in elderly populations, which can exceed 20% in patients older than 65 years [5]. Besides point-of-care diagnostic methods, the discovery of newer anti-TB agents and a more-thorough understanding of its pathophysiology remain important clinical issues in combating this disease.

While the majority of TB studies regarding host immunity have focused on monocytes and lymphocytes, several recent basic science studies have highlighted the role of platelets as a novel participant in TB’s pathogenesis [6]. It was first observed that platelet activation status was correlated with TB disease severity [7]. In active TB, a hypercoagulable state exists with an increased risk of thrombosis, which improves after treatment [8]. Furthermore, it is now recognized that platelets play a role in tissue destruction and pro-inflammation in TB disease [9]. In a recent study, active platelets were present at sites of pulmonary TB, and a co-culture with platelets decreased the intracellular killing of *Mycobacterium tuberculosis* and increased its replication [9]. Finally, another more recent study further provided in vivo experimental evidence that infection-induced platelet activation is a potential target for TB host directed therapy [10]. With accumulating evidence of platelet involvement in TB’s pathophysiology, it was therefore intriguing to determine if the use of antiplatelet agents would be beneficial for TB patients and could serve as a potential adjunct anti-TB agent. However, no epidemiologic studies have examined this question.

We therefore initiated this population-based study to investigate the clinical impacts of antiplatelet agents on the survival of TB patients. Through the linkage of the Taiwan National Tuberculosis Registry (TNTR) database, the Taiwan National Health Insurance (NHI) database, and national mortality data, we recruited incident TB cases and evaluated whether antiplatelet agents were associated with better outcomes.

## 2. Experimental Section

### 2.1. Ethics Statement

The Institutional Review Board of Taipei Medical University approved the study (N201712019) and waved the need for informed consent because this retrospective study used encrypted data and presented no risk to participants.

### 2.2. Study Participants and Setting

This study was conducted by linking Taiwan NHI claims data, mortality data from the Department of Statistics, Ministry of Health, and Welfare, and the TNTR [11]. The TNTR was established by the Taiwan Centers for Disease Control (CDC) in 1996, and clinicians are obligated to report and register every TB patient in Taiwan in the TNTR [12,13]. In addition, the registry system includes information on TB characteristics, treatment courses, and clinical outcomes. Taiwan’s NHI is a universal healthcare system that covers 96% of the residents of Taiwan (with a population of about 23 million) [14,15,16,17].

The inclusion criterion was incident TB cases who received anti-TB treatment identified from the TNTR between 2008 and 2014. Patients with multidrug-resistant TB (MDRTB), with incomplete data, or who were younger than 20 years were excluded.

### 2.3. Definitions and Data Collection

A diagnosis of TB and information regarding TB disease characteristics (smear positivity, culture positivity, and a cavitation on chest radiography) were ascertained from the TNTR. In Taiwan, a diagnosis of TB is made based on clinical symptoms, microbiological studies, radiographic findings, and response to anti-TB treatment [18]. Comorbidities and clinical characteristics of TB patients were extracted from the Taiwanese NHI claims database.

Antiplatelets were divided into aspirin (ASA, irreversible cyclooxygenase inhibitor) and non-ASA antiplatelets, including adenosine diphosphate (ADP) receptor inhibitors, phosphodiesterase inhibitors, glycoprotein IIB/IIIA inhibitors, and adenosine reuptake inhibitors (Appendix A). Protease-activated receptor (PAR)-1 antagonists and thromboxane receptor antagonists were not available in Taiwan during the study period and were therefore excluded from our study.

Users of each category of drugs were defined as using more than 90 defined daily doses (DDDs) of all drugs in the category within 180 days prior to the TB diagnosis. The calculation of DDD followed its definition by the WHO, which is the assumed average maintenance dose per day for a drug used for its main indication in adults [19].

We collected information (DDD) regarding the usage of statins, metformin, nonsteroidal anti-inflammatory drugs (NSAIDs), and corticosteroids within 180 days prior to the TB diagnosis. 

The definition of comorbidities is summarized in Appendix A. Immunocompromised hosts were defined if they had either diabetes mellitus (DM), end-stage renal disease (ESRD), cancer, cirrhosis of the liver, steroid use, a transplant, or acquired immunodeficiency syndrome (AIDS).

### 2.4. Outcomes

The primary outcome was patient survival after a TB diagnosis. Secondary outcomes were mortality within 12 months after the diagnosis and major bleeding events after the diagnosis.

The definition of major bleeding was modified from the International Society on Thrombosis and Haemostasis (ISTH) definition, and modifications were made due to the limitations of the claims database [20,21]. The definition of major bleeding in our study was hospitalization after TB diagnosis due to either an intracranial hemorrhage or a gastrointestinal hemorrhage necessitating a transfusion. The International Classification of Diseases, Ninth Revision, Clinical Modification (ICD-9-CM) and ICD-10-CM codes for defining major bleeding are described in Appendix A. The date of hospitalization was the bleeding date.

All participants were followed up until end of the study period (31 December 2016).

### 2.5. Statistical Analysis

Proportions or means were used to describe the demographic, clinical, and radiographic characteristics of TB patients. Inter-group differences were analyzed using an independent-sample *t*-test for continuous variables and a chi-squared test for categorical variables. The propensity score (PS) for the probability of being administered an antiplatelet agent was derived using a logistic regression model including the potential confounders of age, sex, socioeconomic status, smear positivity, culture positivity, cavitation, comorbidities, the Charlson comorbidity index (CCI) [22], coronary artery disease, DM, ESRD, cancer, cirrhosis, ischemic stroke, chronic obstructive pulmonary disease, AIDS, and hypertension.

A Cox proportional hazard regression model and logistic regression were, respectively, used to analyze factors associated with patient survival and one-year mortality. PS-matching and PS-stratified analysis were both performed. A cumulative incidence function was used to analyze major bleeding events due to competing risks. Variables that remained significantly different after PS matching and accounting for concomitant medications, including statins, steroids, metformin, and NSAIDs, were further adjusted in the final model. We also performed survival and one-year mortality rate analyses before PS matching, adjusted with variables used in the PS derivation. Subgroup analyses were performed among immunocompromised TB patients, including DM, ESRD, psoriasis, cancer, and steroids users. All data analyses were performed using SAS version 9.4 (SAS Institute, Cary, NC, USA). *P* < 0.05 of a two-sided test was considered statistically significant.

## 3. Results

### 3.1. Patients Identification

The patient recruitment process is illustrated in Figure 1. In total, 74,753 participants were recruited for the study.

### 3.2. Demographic Data

The clinical characteristics of identified participants are described in Table 1. Among 74,753 participants, 12.7% (*n* = 9497) were antiplatelet users, and 10.4% (*n* = 7764) were aspirin users. The mean age was 63.6 years for all participants, and 69.8% were male. Compared to non-users, antiplatelet users were older, more likely to be male, and had higher CCI scores. Antiplatelet users were also more likely to have underlying comorbidities, except for AIDS, rheumatoid arthritis (RA), transplant, and pneumoconiosis. Antiplatelet users were also more likely to be culture positive, but they were less likely to be smear positive or have a cavitation on chest radiography. Antiplatelet users also took a higher cumulative dose of statins, NSAIDs, metformin, and corticosteroids (Appendix A).

After PS matching, 8864 antiplatelet users, including 7281 ASA users and 1704 non-ASA antiplatelet users, were matched with 8864 antiplatelet non-users. As for comorbidities, only the stroke incidence significantly differed between the two groups (26.9% versus 24.5%, *p* < 0.001). Antiplatelet users also took higher cumulative doses of statins and metformin (Appendix A).

For the subgroup analysis, a 1:1 comparison cohort was created for the ASA (7281 ASA users and 7281 matched antiplatelet non-users) and non-ASA antiplatelet (1704 non-ASA antiplatelet users and 1704 matched antiplatelet non-users) groups from the PS-matched population (Table 2). For the ASA subgroup analysis, only the stroke incidence significantly differed between ASA users and matched non-users (24.7% versus 22.6%, *p* = 0.003). ASA users also took higher cumulative doses of statins and metformin and a lower dose of corticosteroids compared to non-users (Appendix A). For the non-ASA antiplatelet subgroup analysis, only ESRD significantly differed between non-ASA antiplatelet users and matched non-users (10.9% versus 8.0%, *p* = 0.005) (Table 2). Non-ASA antiplatelet users also took a higher cumulative dose of statins compared to non-users (Appendix A).

### 3.3. Survival Analysis

Before PS matching, antiplatelet use was associated with worse overall survival in the univariate analysis (crude hazard ratio (HR): 1.88, 95% confidence interval (CI): 1.82–1.94, *p* < 0.0001) but better overall survival in the multivariate analysis (adjusted HR: 0.90, 95% CI: 0.86–0.93, *p* < 0.0001). After PS matching, antiplatelet use was associated with improved overall survival in the univariate (crude HR: 0.90, 95% CI: 0.87–0.93, *p* < 0.0001) and multivariate analyses (adjusted HR: 0.91, 95% CI: 0.88–0.95, *p* < 0.0001). In the PS-stratified analysis, antiplatelet use was associated with improved overall survival (adjusted HR: 0.91, 95% CI: 0.84–0.97, *p* < 0.0001).

After PS matching, non-ASA antiplatelet use was associated with worse overall survival compared with ASA use (adjusted HR: 1.36, 95% CI: 1.26–1.46, *p* < 0.0001).

As for the comparison of survival between ASA users and antiplatelet non-users, ASA use was associated with improved overall survival (adjusted HR: 0.90, 95% CI: 0.86–0.94, *p* < 0.0001) after PS matching.

As for the comparison of survival between non-ASA antiplatelet users and antiplatelet non-users, there was no association between non-ASA antiplatelet use and overall survival (HR: 1.00, 95% CI: 0.92–1.09, *p* = 0.996) after PS matching.

In the subgroup analysis, while immunocompromised patients had worse survival compared with immunocompetent patients (adjusted HR: 1.47, 95% CI: 1.41–1.54, *p* < 0.0001), antiplatelet use was associated with better survival among both immunocompromised patients (HR: 0.93, 95% CI: 0.89–0.98, *p* = 0.010) and immunocompetent patients (HR: 0.92, 95% CI: 0.86–0.97, *p* = 0.003). Results of the subgroup analysis are illustrated in Figure 2.

### 3.4. One-Year Mortality Analysis

Before PS matching, antiplatelet use was associated with a higher one-year mortality rate (crude OR: 1.93, 95% CI: 1.83–2.03, *p* < 0.0001) in the univariate analysis but a lower one-year mortality rate (adjusted OR: 0.91, 95% CI: 0.85–0.96, *p* = 0.0015) in the multivariate analysis. In the PS-stratified analysis, antiplatelet use was also associated with a lower one-year mortality rate (adjusted OR: 0.90, 95% CI: 0.86–0.93, *p* < 0.0001).

After PS matching, antiplatelet use was also associated with a lower one-year mortality rate in the univariate analysis (crude OR: 0.87, 95% CI: 0.82–0.93, *p* < 0.0001) and in the multivariate analysis (adjusted OR: 0.91, 95% CI: 0.85–0.97, *p* = 0.004).

After PS matching, non-ASA antiplatelet use was associated with a higher one-year mortality rate compared with ASA use (adjusted OR: 1.49, 95% CI: 1.32–1.69, *p* < 0.0001).

As for the comparison of survival between ASA users and antiplatelet non-users, ASA use was associated with a lower one-year mortality rate (adjusted OR: 0.90, 95% CI: 0.83–0.97, *p* < 0.0001) after PS matching.

As for the comparison of survival between non-ASA antiplatelet users and antiplatelet non-users, there was no association between non-ASA antiplatelet use and the one-year mortality rate (OR: 0.94, 95% CI: 0.82–1.09, *p* = 0.439) after PS matching.

### 3.5. Major Bleeding Event Analysis

Incidences of major bleeding were 0.038, 0.035, 0.052, and 0.019 per person-year in antiplatelet users, ASA users, non-ASA antiplatelet users, and antiplatelet non-users, respectively, in the study population before PS matching.

Incidences of major bleeding were 0.037, 0.035, 0.052, and 0.040 per person-year in antiplatelet users, ASA users, non-ASA antiplatelet users, and antiplatelet non-users, respectively, in the study population after PS matching. There was no difference in the bleeding risk between antiplatelet users and non-users in the cumulative incidence function analysis (subhazard ratio: 0.98, 95% CI: 0.90–1.06, *p* = 0.604).

## 4. Discussion

In our population-based study, we found that antiplatelet use was associated with better overall survival and lower 12-month mortality in pulmonary TB patients who received anti-TB treatment. ASA, which constitutes the majority of antiplatelet agents, provided a survival advantage for TB patients and may be a candidate for auxiliary anti-TB treatment. The safety profile of bleeding events was tolerable among antiplatelet users.

The beneficial effects of antiplatelet agents may result from several mechanisms. First, platelets are known to significantly upregulate monocyte matrix metalloproteinase (MMP)-1 expression, which is associated with lung tissue destruction [10]. In another study, ASA also reduced MMP levels in diabetic rats with induced coronary ischemia [23]. Second, platelets may also act through the modulation of monocyte-derived chemokine and inflammasome activation, leading to a phenotype associated with increased bacterial growth [9]. The increased lung destruction and unrestricted bacterial growth due to platelet activation then leads to poor treatment responses and clinical outcomes. In another murine model study, a low-dose of ASA in combination with other anti-TB agents had anti-inflammatory effects and demonstrated a systemic decrease in neutrophilic recruitment and decreased levels of acute-phase reaction cytokines in the late stage [24]. Interestingly, our study also found that TB disease among antiplatelet users tended not to be cavitated on chest radiography and was also smear negative. This supports antiplatelet agents having beneficial effects of controlling bacterial growth and attenuating lung tissue destruction.

Antiplatelet agents prevent platelet aggregation and are widely used in patients with coronary heart diseases and cerebrovascular events [25]. Different antiplatelet agents inhibit platelet aggregation through different mechanisms. ASA, the most common antiplatelet agent, mainly irreversibly blocks the enzyme cyclooxygenase (COX)-1, thereby preventing the generation of thromboxane A2, a potent platelet activator [26]. Adenosine diphosphate (ADP) antagonists, e.g., clopidogrel, inhibit platelet aggregation by interacting with two purinergic receptors on platelets [27]. A survival advantage was observed in ASA but not in non-ASA antiplatelets. Recently, purinergic pathways were implicated in augmenting the response of killing intracellular pathogens, including *M. tuberculosis,* and also controlling collateral damage [28]. Inhibiting purinergic pathways, as influenced by ADP receptor antagonists, may have adverse impacts on TB treatment. This may explain why a survival benefit was not observed in non-ASA antiplatelet groups. Another explanation may be that non-ASA antiplatelet agents are usually second-line antiplatelets, indicating that non-ASA antiplatelet users may be more disabled or morbid, thus diminishing the benefits of antiplatelet agents. In addition, interestingly, while some basic scientific studies proposed the benefits of ASA, NSAIDs, and even cilostazol in TB treatment, there are no such studies so far that highlighted ADP receptor antagonists as a potential adjunct anti-TB treatment [29,30,31].

Another important and potential explanation for the finding that overall survival did not improve in non-ASA antiplatelet users when compared to antiplatelet non-users may be that the survival benefit was specific to ASA or NSAID effects but not to anti-platelet effects. ASA can modulate inflammatory processes, such as neutrophil attraction and activation, through small lipid mediators such as thromboxane A2 and prostaglandin E2 through COX pathways [32,33]. Interestingly, NSAIDs have been proposed as an adjunct anti-TB treatment by modulating neutrophil recruitment [34,35]. ASA has also been shown before to reduce neutrophil recruitment in active TB diseases [24]. The abovementioned mechanism may also explain why ASA but not non-ASA antiplatelets were associated with improvement in TB survival.

There have been examples of antiplatelet agents being repurposed for other diseases [26,36,37]. An important instance is the use of antiplatelets in chemoprevention and the treatment of colorectal cancer. It was hypothesized that activated platelets contribute to colorectal tumorigenesis and metastatization via cell–cell interactions and release of tumor mediators [26,36]. A low-dose of ASA therefore prevents tumor growth and metastasis by inhibiting platelet activation during different stages of intestinal tumorigenesis [26,36]. As ASA is an old drug with a well-known safety profile, so repurposing ASA use in anti-TB treatment is an attractive approach to improve TB control.

The TNTR is a web-based national TB notification system maintained by the Taiwanese CDC. Since reporting suspected and confirmed TB diseases to the CDC is mandatory and demanded by law in Taiwan, the completeness and timeliness of the TNTR are excellent. Additional patient and disease characteristics, including smear and culture positivity, are available in the database [38]. Uncertain diagnoses of patients also undergo discussion in an expert meeting to ensure the quality of the diagnoses. Linkage between the TNTR, NHI claims database, and mortality data therefore offered the advantage of high diagnostic accuracy, data abundance, and longitudinal follow-ups [12,39].

Bleeding is a concern with use of antiplatelet agents. A low-dose ASA is defined as 75–325 mg daily, and this is the most widely prescribed dosage of ASA used in cardiovascular diseases, but it still carries an increased risk of bleeding [40]. While the vast majority of ASA tablets in Taiwan contain 75–100 mg of ASA, using a DDD of 90 within 180 days as a cutoff point correlated to a daily dose of around 50–100 mg in our study. The bleeding risk in antiplatelet users was not increased in our study, while the survival benefit remained. A low-dose of ASA may, therefore, be considered if ASA is to be used as an auxiliary therapy for TB.

Other concerns remain regarding the use of antiplatelets in anti-TB regimens in addition to the bleeding risk. Drug–drug interactions may also be another issue. For instance, ASA demonstrated a modest antagonism against isoniazid [41,42]. Rifampicin increases the metabolite of clopidogrel, augments antiplatelet activity, and could be associated with a higher bleeding risk [43]. These safety and therapeutic issues should be taken into consideration and evaluated in future studies.

In our study, the benefit of antiplatelet agents was evident among DM, ESRD, psoriasis, and immunocompromised patients. Indeed, diabetes is associated with increased platelet reactivity, and antiplatelet agents are commonly used for the primary and secondary prevention of cardiovascular events [44,45]. ESRD is also associated with the prothrombotic status of platelet dysfunction [46]. In psoriatic patients, an enhanced cyclooxygenase activity with platelelet hyper-aggregation has been observed [47]. In DM, psoriasis, and ESRD patients who develop active TB, the prothrombotic state may be aggravated. In immunocompromised hosts, delayed and impaired innate immune responses to invasion by *M. tuberculosis* are associated with disease status progression, as in the case of DM [48,49]. Furthermore, the treatment of TB patients with underlying comorbidities is a challenge, and treatment outcomes are usually worse. The beneficial effects of antiplatelets in attenuating thrombosis and immunomodulation in DM, psoriasis, ESRD, and immunocompromised patients may be more evident and worthwhile.

Our study has some limitations. First, we were unable to exclude the possibility of unmeasured confounders. For instance, antiplatelet users may have a higher socioeconomic status (Table 1) and tend to be more health conscious. This may lead to the earlier detection of TB and better treatment outcomes. We, however, included disease severity and a low income in our PS matching. Furthermore, while antiplatelet agents are mainly used for the primary and secondary prevention of cardiovascular and cerebrovascular diseases, which are usually the result of end organ damage of underlying diseases and aging processes, antiplatelet users may be more fragile and disabled than non-users. Second, our study was conducted in Taiwan, the population of which is of Asian ethnicity. Whether these findings can be extrapolated to patients of other ethnicities remains unknown. In addition, this was a retrospective study, and the conclusions drawn from our study can be speculative. Future interventional studies, especially with a randomized design, should be conducted to prove this finding.

In conclusion, our study is the first population-based epidemiological study that has demonstrated a survival benefit among active TB patients receiving antiplatelets. Our study points to a potential direction for designing and discovering new anti-TB agents. Since ASA use was associated with better overall survival and other antiplatelet drugs did not improve survival rates, this phenomenon may suggest that the survival benefit may not be solely due to antiplatelet effects. Additional studies investigating the underlying mechanisms of antiplatelet treatment against TB and co-morbidities would also be of interest.

## 5. Data Availability

All data are deposited in Department of Statistics, Ministry of Health and Welfare, Taiwan and are not allowed to exported without application and permission.

## Figures and Tables

**Figure 1 jcm-08-00923-f001:**
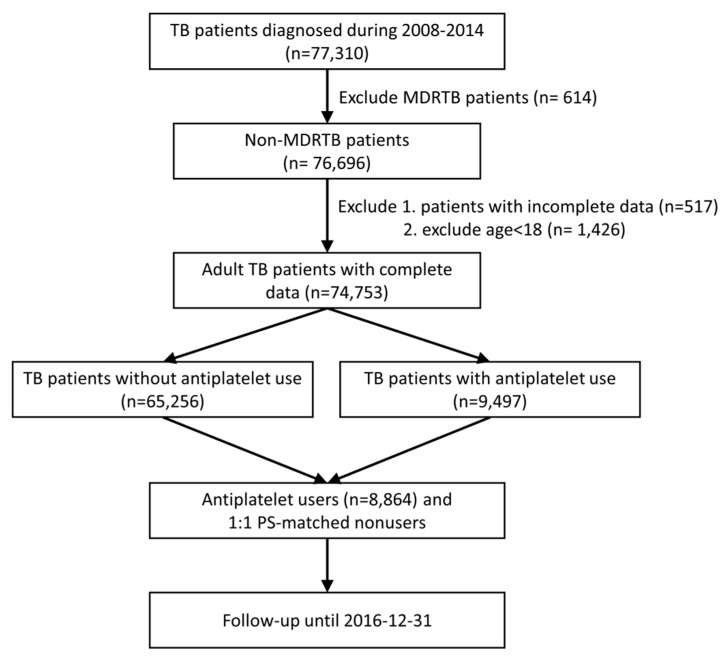
Flowchart of patient recruitment.

**Figure 2 jcm-08-00923-f002:**
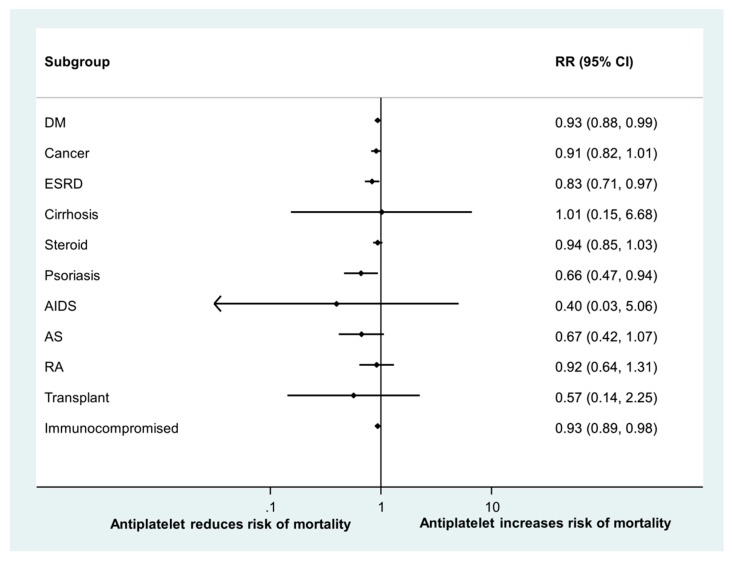
Forest plot of the association between antiplatelet agent use and survival among different groups of patients. Abbreviations: AIDS, acquired immunodeficiency syndrome; AS, ankylosing spondylitis; DM, diabetes mellitus; ESRD, end-stage renal disease; and RA, rheumatoid arthritis

**Table 1 jcm-08-00923-t001:** Clinical characteristics of tuberculosis patients with and those without antiplatelet use.

	Before PS matching						After PS matching					
	Overall (*N* = 74,753)	Antiplatelet users (N = 9497)			Non-users (*n* = 65,256)	*p* value *	Overall (*n* = 17,728)	Antiplatelet users (*N* = 8864)			Non-users (*n* = 8864)	*p* value *
		All (*n* = 9497)	ASA (*n* = 7764)	Non-ASA (*n* = 1855)				All (*n* = 8864)	ASA (*n* = 7281)	Non-ASA (*n* = 1704)		
**Age (mean ± SD)**	63.6 ± 18.5	76.1 ± 10.6	75.7 ± 10.7	77.5 ± 9.9	61.7 ± 19.7	< 0.001	75.9 ± 10.6	75.7 ± 10.6	75.4 ± 10.7	77.2 ± 9.9	76.0 ± 10.6	0.063
**Male sex**	52,155 (69.8%)	7019 (73.9%)	5702 (73.4%)	1409 (76.0%)	45,136 (69.2%)	< 0.001	13,059 (73.7%)	6568 (74.1%)	5368 (73.7%)	1295 (76.0%)	6,491 (73.2%)	0.195
**Low income**	4,005 (5.4%)	303 (3.2%)	250 (3.2%)	59 (3.2%)	3702 (5.7%)	< 0.001	529 (3.0%)	286 (3.2%)	234 (3.2%)	47 (2.8%)	243 (2.7%)	0.064
**CCI (mean ± SD)**	3.7 ± 2.1	5.1 ± 2.0	5.0 ± 2.0	5.6 ± 2.1	3.5 ± 2.6	< 0.001	5.1 ± 2.0	5.1 ± 2.0	4.9 ± 2.0	5.5 ± 2.1	5.1 ± 2.0	0.804
**CAD**	20,975 (28.1%)	6,571 (69.2%)	5263 (67.8%)	1444 (77.8%)	14,404 (22.1%)	< 0.001	12,303 (69.4%)	6097 (68.8%)	4911 (67.5%)	1,319 (77.4%)	6206 (70.0%)	0.078
**Stroke**	6313 (8.5%)	2610 (27.5%)	1968 (25.4%)	655 (35.3%)	3703 (5.7%)	< 0.001	4550 (25.7%)	2381 (26.9%)	1799 (24.7%)	599 (35.2%)	2169 (24.5%)	< 0.001
**DM**	14,779 (19.8%)	3604 (38.0%)	2950 (38.0%)	717 (38.7%)	11,175 (17.1%)	< 0.001	6666 (37.6%)	3365 (38.0%)	2775 (38.1%)	654 (38.4%)	3301 (37.2%)	0.329
**ESRD**	1911 (2.6%)	500 (5.3%)	325 (4.2%)	207 (11.2%)	1411 (2.2%)	< 0.001	885 (5.0%)	451 (5.1%)	293 (4.0%)	185 (10.9%)	434 (4.9%)	0.581
**Cancer**	7730 (10.3%)	1086 (11.4%)	839 (10.8%)	253 (13.6%)	6,644 (10.2%)	< 0.001	1969 (11.1%)	993 (11.2%)	773 (10.6%)	231 (13.6%)	976 (11.0%)	0.702
**AIDS**	491 (0.7%)	7 (0.1%)	7 (0.1%)	0 (0%)	484 (0.7%)	< 0.001	10 (0.1%)	7 (0.1%)	7 (0.1%)	0 (0%)	3 (0.03%)	0.343
**RA**	1004 (1.3%)	127 (1.3%)	101 (1.3%)	33 (1.8%)	877 (1.3%)	0.996	242 (1.4%)	118 (1.3%)	95 (1.3%)	30 (1.8%)	124 (1.4%)	0.746
**Psoriasis**	705 (0.9%)	114 (1.2%)	78 (1.0%)	36 (1.9%)	591 (0.9%)	0.007	211 (1.2%)	104 (1.2%)	73 (1.0%)	30 (1.8%)	107 (1.2%)	0.890
**AS**	613 (0.8%)	97 (1.0%)	75 (1.0%)	22 (1.2%)	516 (0.8%)	0.023	173 (1.0%)	92 (1.0%)	70 (1.0%)	22 (1.3%)	81 (0.9%)	0.445
**COPD**	13,187 (17.6%)	2660 (28.0%)	2135 (27.5%)	568 (30.6%)	10,527 (16.1%)	< 0.001	5045 (28.5%)	2,503 (28.2%)	2,014 (27.7%)	535 (31.4%)	2,542 (28.7%)	0.527
**Transplant**	135 (0.2%)	24 (0.3%)	17 (0.2%)	8 (0.4%)	111 (0.2%)	0.101	40 (0.2%)	24 (0.3%)	17 (0.2%)	8 (0.5%)	16 (0.2%)	0.268
**Pneumoconiosis**	69 (0.1%)	13 (0.1%)	11 (0.1%)	**	56 (0.1%)	NS	20 (0.1%)	11 (0.1%)	9 (0.1%)	**	9 (0.1%)	NS
**Bronchiectasis**	1474 (2.0%)	233 (2.5%)	192 (2.5%)	46 (2.5%)	1241 (1.9%)	< 0.001	462 (2.6%)	216 (2.4%)	177 (2.4%)	42 (2.5%)	246 (2.8%)	0.172
**Hypertension**	37867 (50.7%)	8688 (91.5%)	7095 (91.4%)	1717 (92.6%)	29179 (44.7%)	< 0.001	16259 (91.7%)	8095 (91.3%)	6646 (91.3%)	1572 (92.2%)	8164 (92.1%)	0.064
**TB severity**												
**Smear positive**	30,257 (40.5%)	3445 (36.3%)	2873 (37.0%)	621 (33.5%)	26,812 (41.1%)	< 0.001	6731 (38.0%)	3348 (37.8%)	2795 (38.4%)	604 (35.6%)	3383 (38.2%)	0.599
**Culture positive**	56,383 (75.4%)	7343 (77.3%)	6030 (77.7%)	1389 (74.9%)	49,040 (75.2%)	< 0.001	13,575 (76.6%)	6784 (76.5%)	5598 (76.9%)	1,263 (74.1%)	6791 (76.6%)	0.915
**Cavitation**	12,092 (16.2%)	950 (10.0%)	832 (10.7%)	120 (6.5%)	11,142 (17.1%)	< 0.001	1832 (10.3%)	927 (10.5%)	816 (11.2%)	114 (6.7%)	905 (10.2%)	0.604

Abbreviations: AIDS, acquired immunodeficiency syndrome; AS, ankylosing spondylitis; CAD, coronary artery disease; CCI, Charlson comorbidity index; COPD, chronic obstructive pulmonary disease; DM, diabetes mellitus; ESRD, end-stage renal disease; NS, non-significant; PS, propensity score; RA, rheumatoid arthritis; SD, standard deviation; and TB, tuberculosis. * Compared between all antiplatelet users and non-users. ** According to the regulations of National Health Insurance claims database, results with case number less than three are not allowed to be exported.

**Table 2 jcm-08-00923-t002:** Subgroup analysis of aspirin users (ASA), non-aspirin antiplatelet users (non-ASA), and non-antiplatelet users (non-user) after propensity-score matching.

	ASA (*n* = 7281)	Matched non-user (*n* = 7281)	*p* value	Non-ASA (*n* = 1704)	Matched non-user (*n* = 1704)	*p* value
**Age** (mean ± SD)	75.4 ± 10.7	75.7 ± 10.7	0.072	77.2 ± 9.9	77.6 ± 9.8	0.244
**Male sex**	5368 (73.7%)	5317 (73.0%)	0.349	1295 (76.0%)	1275 (74.8%)	0.450
**Low income**	234 (3.2%)	198 (2.7%)	0.087	47 (2.8%)	43(2.5%)	0.749
**CCI**	4.9 ± 2.0	5.0 ± 2.0	0.474	5.5 ± 2.1	5.5 ± 2.1	0.634
Coronary artery disease	4911 (67.5%)	4990 (68.5%)	0.166	1319 (77.4%)	1339 (78.6%)	0.432
Stroke	1799 (24.7%)	1647 (22.6%)	0.003	599 (35.2%)	551 (32.3%)	0.089
Diabetes mellitus	2775 (38.1%)	2715 (37.3%)	0.313	654 (38.4%)	649 (38.1%)	0.888
End-stage renal disease	293 (4.0%)	306 (4.2%)	0.617	185 (10.9%)	136 (8.0%)	0.005
Cancer	773 (10.6%)	778 (10.7%)	0.914	231 (13.6%)	210 (12.3%)	0.307
AIDS	7 (0.1%)	3 (0.04%)	0.343	0 (0%)	0 (0%)	> 0.999
Rheumatoid arthritis	95 (1.3%)	96 (1.3%)	> 0.999	30 (1.8%)	27 (1.6%)	0.789
Psoriasis	73 (1.0%)	81 (1.1%)	0.571	30 (1.8%)	26 (1.5%)	0.686
Ankylosing spondylitis	70 (1.0%)	66 (0.9%)	0.796	22 (1.3%)	15 (0.9%)	0.321
COPD	2014 (27.7%)	2051 (28.2%)	0.506	535 (31.4%)	533 (31.3%)	0.971
Transplant	17 (0.2%)	14 (0.2%)	0.719	8 (0.5%)	5 (0.3%)	0.578
Pneumoconiosis	9 (0.1%)	8 (0.1%)	> 0.999	**	**	NS
Bronchiectasis	177 (2.4%)	203 (2.8%)	0.194	42 (2.5%)	47 (2.8%)	0.668
**Tuberculosis severity**						
Smear positive	2795 (38.4%)	2829 (38.9%)	0.574	604 (35.6%)	609 (35.8%)	0.886
Culture positive	5598 (76.9%)	5602 (76.9%)	0.953	1263 (74.1%)	1272 (74.7%)	0.754
Cavitation	816 (11.2%)	785 (10.8%)	0.427	114 (6.7%)	120 (7.0%)	0.735

Abbreviations: AIDS, acquired immunodeficiency syndrome; COPD, chronic obstructive pulmonary disease; CCI, Charlson comorbidity index; NS, non-significant; and SD, standard deviation. * Compared between all antiplatelet users and non-users. ** According to the regulations of National Health Insurance database, results with case number less than three are not allowed to be exported.

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
