# Peer review of "Use of Antiplatelet Agents and Survival of Tuberculosis Patients: A Population-Based Cohort Study"

_jcm, 2019, doi:10.3390/jcm8070923_

Round 1

Reviewer 1 Report

Lee et al. provide interesting findings indicating the potential beneficial use of anti-platelet agents in addition to anti-TB therapy. Overall, the authors provide sound experimental design to address their research question. Additional studies investigating the mechanisms of anti-platelet treatments against TB and co-morbidities are likely to yield interesting findings, given the data presented in this article. 

Author Response

Lee et al. provide interesting findings indicating the potential beneficial use of anti-platelet agents in addition to anti-TB therapy. Overall, the authors provide sound experimental design to address their research question. Additional studies investigating the mechanisms of anti-platelet treatments against TB and co-morbidities are likely to yield interesting findings, given the data presented in this article.

Ans: Thanks for the encouragement. We have also included this in our conclusion section.

Additional studies investigating the underlying mechanisms of antiplatelet treatment against TB and co-morbidities would also be of interest.

Reviewer 2 Report

In the present manuscript the authors describe that Tb patients that have been also treated with the NSAID and COX-1 inhibitor Aspirin showed improved survival rates and 1-year mortality rates when compared to patients that have been treated with other anti-platelet drugs than Aspririn or not treated with anti-platelet drugs at all. However, the authors conclude that the increased survival rates in the Aspirin group are due to effects on platelets, although the group of non-Aspirin anti-platelet users did not show any improvement (see below). Thus, this reviewers asks if the authors drew the correct conclusion from their data and if they could please clarify.

Results

·        Line 188: It is interesting that overall survival did not improve in non-ASA antiplatelet users when compared to antiplatelet non-users. This suggests that the improved overall survival rates in ASA users was specific to ASA effects, but not to anti-platelet effects. Put the other way around: Improved survival rates were not due to ASA’s effects on platelets, but due to other ASA effects on other regulatory networks. Same for the 1-year mortality rate (line 211). Aspirin is an NSAID that acts on COX-1 and COX-2, therefore, modulating a variety of small lipid mediators such as thromboxane A2 and PGE2 among others. Those small lipid mediators are not only involved in platelet regulation, but in many other inflammatory processes such as neutrophil attraction and activation. Thus, conclusions like “we found that antiplatelet use was associated with better overall survival“ (line 227 of the discussion) may need to be changed to “ASA use was associated with better overall survival, but not due to anti-platelets effects” since other antiplatelet drugs did not improve survival rates. Could the authors give a comment on that observation and clarify please?

Discussion

The authors mention that ASA has been shown before to reduced neutrophil recruitment (Ref 19). As stated above, the effects the authors find in their study may not be platelet-specific but NSAID-specific. NSAIDs as adjunct Tb treatment have been discussed before (PMID: 23564636; PMID: 29055689).

Minor

·        Section 2.2 may be shortened

·        Section 2.3 may benefit from rephrasing since its convoluted sentences make it hard to understand for the reader during the first read-through

Author Response

Major comments:

In the present manuscript the authors describe that TB patients that have been also treated with the NSAID and COX-1 inhibitor Aspirin showed improved survival rates and 1-year mortality rates when compared to patients that have been treated with other anti-platelet drugs than Aspirin or not treated with anti-platelet drugs at all. However, the authors conclude that the increased survival rates in the Aspirin group are due to effects on platelets, although the group of non-Aspirin anti-platelet users did not show any improvement (see below). Thus, this reviewer asks if the authors drew the correct conclusion from their data and if they could please clarify.

Results

Line 188: It is interesting that overall survival did not improve in non-ASA antiplatelet users when compared to antiplatelet non-users. This suggests that the improved overall survival rates in ASA users was specific to ASA effects, but not to anti-platelet effects. Put the other way around: Improved survival rates were not due to ASA’s effects on platelets, but due to other ASA effects on other regulatory networks. Same for the 1-year mortality rate (line 211). Aspirin is an NSAID that acts on COX-1 and COX-2, therefore, modulating a variety of small lipid mediators such as thromboxane A2 and PGE2 among others. Those small lipid mediators are not only involved in platelet regulation, but in many other inflammatory processes such as neutrophil attraction and activation. Thus, conclusions like “we found that antiplatelet use was associated with better overall survival“ (line 227 of the discussion) may need to be changed to “ASA use was associated with better overall survival, but not due to anti-platelets effects” since other antiplatelet drugs did not improve survival rates. Could the authors give a comment on that observation and clarify please?

Discussion

The authors mention that ASA has been shown before to reduced neutrophil recruitment (Ref 19). As stated above, the effects the authors find in their study may not be platelet-specific but NSAID-specific. NSAIDs as adjunct Tb treatment have been discussed before (PMID: 23564636; PMID: 29055689).

Ans: Thanks for the excellent comments. We have revised the conclusions and added a paragraph in the discussion tackling with this issue.

Another important and potential explanation for the finding that the overall survival did not improve in non-ASA antiplatelet users when compared to antiplatelet non-users may be that the survival benefit was specific to ASA or NSAID effects, but not to anti-platelet effects. ASA can regulate inflammatory processes, such as neutrophil attraction and activation through small lipid mediators such as thromboxane A2 and prostaglandin E2 through COX pathways [32,33]. Interestingly, NSAIDs have been proposed as adjunct anti-TB treatment by modulating neutrophil recruitment [34,35]. ASA has also been shown before to reduce neutrophil recruitment in active TB diseases [24]. The abovementioned mechanism may also explain why ASA but not non-ASA antiplatelet was associated with improvement in TB survival.”

Minor comments

Section 2.2 may be shortened

Ans: Thanks for the suggestions. We have shortened section 2.2 and added some references.

This study was conducted by linking Taiwan NHI claims data, mortality data from the Department of Statistics, Ministry of Health and Welfare, and the TNTR [11]. The TNTR was established by the Taiwan Centers for Disease Control (CDC) in 1996, and clinicians are obligated to report and register every TB patient in Taiwan in the TNTR [12,13]. Also, the registry system includes information on TB characteristics, treatment courses, and clinical outcomes. Taiwan's NHI is a universal healthcare system that covers 96% of the residents of Taiwan (with a population of about 23 million) [14-17].

The inclusion criterion was incident TB cases who received anti-TB treatment identified from the TNTR between 2008 and 2014. Patients with multidrug-resistant TB (MDRTB), with incomplete data, or who were younger than 20 years were excluded.”

Section 2.3 may benefit from rephrasing since its convoluted sentences make it hard to understand for the reader during the first read-through

Ans: Thanks for the suggestions. We have rephrased section 2.3 to make it easier for the readers to follow. We also shifted the definition of steroid user to appendix table 2 to simplify this section. 

“A diagnosis of TB and information regarding TB disease characteristics (smear positivity, culture positivity, and cavitation on chest radiography) were ascertained from the TNTR. In Taiwan, a diagnosis of TB is made based on clinical symptoms, microbiological studies, radiographic findings, and response to anti-TB treatment [18]. Comorbidities and clinical characteristics of TB patients were extracted from the Taiwanese NHI claims database.

Antiplatelets were divided into aspirin (ASA, irreversible cyclooxygenase inhibitor) and non-ASA antiplatelets, including adenosine diphosphate (ADP) receptor inhibitors, phosphodiesterase inhibitors, glycoprotein IIB/IIIA inhibitors, and adenosine reuptake inhibitors (Appendix Table 1). Protease-activated receptor (PAR)-1 antagonists and thromboxane receptor antagonists were not available in Taiwan during the study period and were therefore excluded from our study.

Users of each category of drugs were defined as using more than 90 defined daily doses (DDDs) of all drugs in the category within 180 days prior to the TB diagnosis. The calculation of DDD followed its definition by the WHO, which is the assumed average maintenance dose per day for a drug used for its main indication in adults [19].”

Reviewer 3 Report

This is a very interesting study on overall survival and 1-year mortality in patients affected by tuberculosis (TB) treated and not treated with antiplatelet agents. In this retrospective study the authors included more than 70000 incident TB cases in the Taiwan National Tuberculosis Registry from 2008 to 2014. After a careful statistical analysis they concluded that use of antiplatelet agents in TB patients is associated with an improved survival without any higher bleeding complication than in those not using antiplatelets. Although I admit that this paper is novel and well-written, I have some concerns/suggestions for the authors in order to improve their work.

Major concerns/suggestions

1)      Lines 111-112: Please, use the ISTH definition for major bleeding.

2)      Lines 126-127: Multivariate analyses before PS matching for overall survival and 1-year mortality should include comorbidities, as calculated by the CCI, as covariate.

3)      Lines 185-190: In this paragraph the authors should report only the direct comparison between patients treated with different antiplatelets (ASA vs non-ASA users).

4)      Lines 191-196: Please, delete the statistical analyses between sub-groups (psoriasis???). I’ll keep only the comparison between patients immunocompetent vs immunocompromised.

5)      Line 251 “A survival advantage was observed in ASA but not in non-ASA antiplatelets”: This sentence might be correct, but a direct comparison between patients treated with different antiplatelets (ASA vs non-ASA users) is due.

6)      In the “Limitation” section of your paper the authors should recognize and report the important limitations due to the fact that this study was retrospective. Thus, the conclusions can be hypothesized, but they remain pure speculation. Only a longitudinal study can be considered as definitive.

Author Response

Major concerns/suggestions

1) Lines 111-112: Please, use the ISTH definition for major bleeding.

Ans: Thanks for the suggestions. The definition of major bleeding was modified from ISTH definition and included important components of ISTH definition. Due to the limitations of claims database, we were unable to make it the same with the ISTH definition for major bleeding (for instance, hemoglobin drop is unable to be ascertained from claims database, uncommon sites of bleeding cannot be reliably coded, also surgical and nonsurgical major bleeding were difficult to differentiate. We have revised the description for the definition of major bleeding as following.

The definition of major bleeding was modified from International Society on Thrombosis and Haemostasis (ISTH) definition and modifications were made due to the limitations of claims database [20,21]. The definition of major bleeding in our study was hospitalization after TB diagnosis due to 1. intracranial hemorrhage or 2. gastrointestinal hemorrhage necessitating a transfusion.”

2) Lines 126-127: Multivariate analyses before PS matching for overall survival and 1-year mortality should include comorbidities, as calculated by the CCI, as covariate.

Ans: Thanks for the suggestions. We were sorry that our manuscript was not clear enough. We did include CCI in the multivariate analyses before PS matching for overall survival and 1-year mortality. To make it clear, we have added a sentence explaining this in the statistical analysis section.

“We also performed survival and one-year mortality rate analysis before PS matching, adjusted with variables used in the PS derivation.”

3) Lines 185-190: In this paragraph the authors should report only the direct comparison between patients treated with different antiplatelets (ASA vs non-ASA users).

Ans: Thanks for the excellent comments. We have revised accordingly. We have both revised the survival analysis and one-year mortality analysis.

“After PS matching, non-ASA antiplatelet use was associated with worse overall survival compared with ASA use (adjusted HR: 1.36, 95% CI: 1.26~1.46, p < 0.0001).”

“After PS matching, non-ASA antiplatelet use was associated with higher one-year mortality rate compared with ASA use (adjusted OR: 1.49, 95% CI: 1.32~1.69, p < 0.0001).”

4) Lines 191-196: Please, delete the statistical analyses between sub-groups (psoriasis???). I’ll keep only the comparison between patients immunocompetent vs immunocompromised.

Ans: Thanks for the excellent comments. We have revised accordingly.

“In the subgroup analysis, while immunocompromised patients had worse survival compared with immunocompetent patients (adjusted HR: 1.47, 95% CI: 1.41~1.54, p < 0.0001), antiplatelet use was associated with better survival among both immunocompromised patients (HR: 0.93, 95% CI: 0.89~0.98, p = 0.010) and immunocompetent patients (HR: 0.92, 95% CI: 0.86-0.97, p = 0.003).”

5) Line 251 “A survival advantage was observed in ASA but not in non-ASA antiplatelets”: This sentence might be correct, but a direct comparison between patients treated with different antiplatelets (ASA vs non-ASA users) is due.

Ans: Thanks for the excellent comments. We have added the results of direct comparison between patients treated with different antiplatelets in the results section.

“After PS matching, non-ASA antiplatelet use was associated with worse overall survival compared with ASA use (adjusted HR: 1.36, 95% CI: 1.26~1.46, p < 0.0001).”

“After PS matching, non-ASA antiplatelet use was associated with higher one-year mortality rate compared with ASA use (adjusted OR: 1.49, 95% CI: 1.32~1.69, p < 0.0001).”

6) In the “Limitation” section of your paper the authors should recognize and report the important limitations due to the fact that this study was retrospective. Thus, the conclusions can be hypothesized, but they remain pure speculation. Only a longitudinal study can be considered as definitive.

Ans: Thanks for the reminder. We have added this in the limitation section.

“Also, this is a retrospective study and the conclusions drawn from our study can be speculative. Future interventional studies, especially with a randomized design, should be conducted to prove this finding.”

Round 2

Reviewer 2 Report

The authors adressed all my points.

Reviewer 3 Report

Accepted in present form